# Adapting COVID-19 research infrastructure to capture influenza and RSV alongside SARS-CoV-2 in UK healthcare workers winter 2022/23: Evaluation of the SIREN Winter Pressures pilot study

**Sophie Russell**[ID]°\*, **Katie Munro**[ID]°\*, **Sarah Foulkes, Jonathan Broad, Dominic Sparkes, Ana Atti, Jasmin Islam, Susan Hopkins, Victoria Hall, SIREN study group**¶

SIREN study team, AMR & HCAI Division, United Kingdom Health Security Agency (UKHSA), London, United Kingdom

¶ Membership of the SIREN study group is provided in the Acknowledgements.
☯ These authors contributed equally to this work.
\* sophie.russell@ukhsa.gov.uk (SR); katie.munro@ukhsa.gov.uk (KM)

## Abstract

### Background

In winter 2022, SIREN, a prospective healthcare worker cohort study monitoring SARS-CoV-2, ran a pilot sub-study introducing multiplex PCR testing for SARS-CoV-2, influenza, and RSV to investigate winter pressures. Three pathways were trialled: (A) on-site (at hospital) swabbing for PCR testing, using the local laboratory for testing, (B) on-site swabbing using a UKHSA-commissioned laboratory for testing, and (C) postal swabbing using a UKHSA-commissioned laboratory for testing. Here, we compare pathways in relation to recruitment, testing coverage, participant acceptability, and UKHSA SIREN research team feedback.

### Methods

A mixed methods evaluation using metrics of quality assurance and study fidelity (participant recruitment and retention; multiplex PCR testing timing and coverage), an adapted NIHR 'participant in research' feedback questionnaire, and thematic analysis of a UKHSA SIREN research team workshop.

### Results

With 7,774 participants recruited, target recruitment (N = 7,500) was achieved. Thirty-nine sites took part in the sub-study (4,289 participants). Thirty-three used pathway A (3,713 participants), and six used pathway B (576 participants). 3,485 participants were enrolled into pathway C (27.8% of invitees). The median number of tests per

**Data availability statement:** All relevant data are within the paper and its Supporting Information files.

**Funding:** The SIREN study was funded by the UK Health Security Agency; the UK Department of Health and Social Care with contributions from the governments in Northern Ireland, Wales, and Scotland; the National Institute for Health Research; Health Data Research UK (NIHR200927; HDRUK2022.0322). The funders had no role in study design, data collection and analysis, decision to publish, or preparation of the manuscript.

**Competing interests:** The authors have declared that no competing interests exist.

participant was similar across pathways (6; 4; 5). However, sites using local laboratories showed a wide variation in the date they switched to multiplex testing (28th November 2022–16th March 2023). Consequently, influenza and RSV testing coverage was higher for pathways using UKHSA-commissioned laboratories (100.0% vs 45.6% at local laboratories). 1,204/7,774 (15.5%) participants completed the feedback survey. All pathways were acceptable to participants; 98.9% of postal and 97.5% of site-based participants 'would consider taking part again'.

## Conclusion

Transitioning SARS-CoV-2 PCR testing to include influenza and RSV was challenging to achieve rapidly across multiple sites. The postal testing pathway proved more agile, and UKHSA-commissioned laboratory testing provided more comprehensive data collection than local laboratory testing. This sub-study indicates that postal protocols are effective, adaptable at pace, and acceptable to participants.

---

## Introduction

This paper presents a process evaluation of the SARS-CoV-2 Immunity and Reinfection EvaluatioN (SIREN) Winter Pressures 2022−23 sub-study. The sub-study was a pilot set up to investigate the compounding burden of three respiratory infections (SARS-CoV-2; influenza; RSV) in healthcare workers (HCWs) [1]. Here, a process evaluation framework is used with the aim of comparing three testing pathways used in the sub-study to determine acceptability to participants, recruitment and retention, testing coverage performance, and delivery across pathways.

### Background to the SIREN Winter Pressures 2022−23 sub-study

In the UK, winter 2022−23 was anticipated to be particularly challenging. Circulating respiratory viruses were predicted to increase due to relaxation of non-pharmaceutical interventions put in place to reduce COVID-19 transmission (e.g., social distancing; masking). This effect had been seen in Australasian data where there was an increase in influenza cases, hospitalisations, and deaths [2,3]. The SIREN study, led by the UK Health Security Agency (UKHSA), was well-placed to adapt its existing infrastructure set up during the COVID-19 pandemic to include multiplex Polymerase Chain Reaction (PCR) testing for influenza and RSV, in addition to SARS-CoV-2.

The Winter Pressures sub-study was a longitudinal cohort study of HCWs nested within the SIREN study running from 28 November 2022 to 31 March 2023. The overall design and methods of the Winter Pressures sub-study, and the wider SIREN study, have been described elsewhere [1,4]. Expanding the study to incorporate multiplex testing required rapid adjustment of the protocol. In the original SIREN study, participants attended hospital sites to undergo fortnightly SARS-CoV-2 PCR testing and completed an online fortnightly questionnaire on symptoms, exposures, and sick days.

The Winter Pressures sub-study aimed to determine the incidence and impact of SARS-CoV-2, influenza, and RSV on the healthcare workforce using multiplex testing. The secondary objective was to evaluate the effectiveness of seasonal influenza vaccination against infection, and power calculations required a sample size of 7,500 participants testing fortnightly to achieve this [1].

Participant recruitment to the sub-study proved challenging to achieve at pace, with sites facing a range of barriers including site study team capacity, laboratory availability, and access to multiplex machines. To increase the likelihood of meeting participant recruitment targets, a new, centrally managed testing pathway was established for participants to undertake PCR swabs via postal kits, in parallel to existing testing pathways at hospital sites.

The sub-study comprised three PCR testing pathways: Pathway A: fortnightly swabs were taken and tested locally at hospital sites. Pathway B: fortnightly swabs were taken locally at hospital sites but sent to UKHSA-commissioned laboratories for testing. Pathway C: fortnightly swabs were sent to participants via the post to perform at home and returned to UKHSA-commissioned laboratories for testing.

### Process evaluation of the SIREN Winter Pressures 2022−23 sub-study

Home-based methodologies similar to Pathway C are growing in popularity for being resource-effective, reducing research costs, and reflecting a participant-centred design [5,6]. However, existing research lacks comparable information on key metrics, and does not address HCW studies where the site is also their place of work [7]. Here, a process evaluation framework has been chosen as an appropriate method to compare site-based and postal testing pathways.

This process evaluation aims to evaluate site-based and postal testing pathways by comparing recruitment, multiplex PCR testing coverage, participant experience and acceptability, and study delivery via UKHSA SIREN research team feedback. This will inform future research protocol design, enabling comparison of postal and site-based testing options.

## Methods

We conducted a process evaluation of the SIREN Winter Pressures sub-study using mixed methods, to compare the performance and acceptability of site-based and postal testing.

We used the following quantitative and qualitative data:

1. Metrics of quality assurance and study fidelity (participant recruitment and retention, timing of introduction and coverage of multiplex PCR testing)

2. Participants research experience survey

3. UKHSA SIREN research team workshop

The methods here are informed by guidance on process evaluation from the Medical Research Council [8].

### Metrics of quality assurance and study fidelity

**Recruitment and retention.** In November 2022, all 65 SIREN sites that intended to offer participant testing until March 2023 were invited to join the SIREN Winter Pressures sub-study. Sites were asked to opt-in to the sub-study via an online survey. Sites were recruited between 18 November 2022 and 21 February 2023.

Sites could choose to conduct PCR swabbing and multiplex testing locally (Pathway A) or multiplex testing via UKHSA-commissioned laboratories (Pathway B). Participants at sites who joined the sub-study were sent an updated Participant Information Leaflet (PIL) informing them of the switch to multiplex testing, and were given the opportunity to withdraw from the study if they did not wish to continue. Consent was implied by continuing to undergo testing. Where possible, participants who had previously been enrolled in the study but were inactive (due to completing their period of follow-up) were re-recruited and consented to testing within the sub-study via online survey (between 09 December 2022 and 15 March 2023).

Participants from sites not taking part in the SIREN Winter Pressures sub-study were re-recruited and consented to join via survey into a postal pathway (Pathway C), between 13 December 2022 and 24 January 2023. This pathway did not require on-site swabbing and testing.

For Pathways A and B, we recorded the number of sites joining the sub-study and the number of participants enrolled at each site when the sub-study went live.

For Pathway C, we recorded the number of participants consented and the date they consented.

To withdraw from the study, participants were required to complete a withdrawal survey. We describe recruitment, retention, and reason for withdrawal by site-based versus postal swabbing. A Fisher's exact test was used to compare withdrawal reasons by pathway.

**Timing of introduction and coverage of multiplex PCR testing.** For sites in Pathway A, we recorded the date they switched to multiplex testing. Where this was unavailable, the date the PILs were sent to their participants was used (as a proxy for the switch to multiplex testing).

For sites in Pathway B, the date the PILs were sent to their participants was used as the date swabs should be sent to UKHSA-commissioned laboratories for multiplex testing.

For participants in Pathway C, we recorded the date the postal PCR swabs were taken.

For all pathways, we calculated swab return rate as the proportion of participants who had returned at least one swab for PCR testing, and calculated the median number of tests per participant. We described the number of monoplex (SARS-CoV-2 only) PCR results and the number of multiplex (SARS-CoV-2 plus influenza and/or RSV) PCR results over time, both overall and by pathway. A chi-squared test was used to compare swab return rates, and to compare the proportion of tests that were multiplex, between pathways.

## Participant survey

A bespoke adaptation of the standardised national NIHR participant research experience survey was created for participants [9]. A copy of the participant survey used can be found in the supplementary information (S1 File). This was an 11-item survey that included quantitative and qualitative assessments based on participant experience. The quantitative elements included participants' views on working with the UKHSA and site SIREN teams, the accessibility of testing, communication, and overall experience, and, for those on the postal pathway, whether they would recommend their pathway. Participants rated their views based on a 5-point Likert scale from strongly agree to strongly disagree. The survey included two free text boxes that encouraged participants to provide positive feedback or suggested improvements without a character limit.

Data were collated via Snapsurvey, an online survey tool, and stored securely on UKHSA internal servers. This questionnaire was sent to all participants who completed the SIREN Winter Pressures sub-study. Survey responses were anonymous. Participants were required to select their swabbing pathway within the survey to enable a comparison of participant experience between site-based and postal study designs. Site-based pathways (Pathways A and B) were grouped for this analysis as the participant experience for both pathways involved site-based swabbing. A comparison of responses was made between site-based and postal pathways using a Fisher's exact test.

## UKHSA SIREN research team workshop

All researchers in the SIREN team were invited to attend a workshop to aid evaluation of the Winter Pressures sub-study. Feedback on the process of study set up and delivery was collected, and organised around core workstreams (e.g., laboratory, ethics, data collection, communications). Feedback was written up into long-form minutes that were thematically analysed.

## Thematic analysis

All free text questions from the participant survey and SIREN research team workshop feedback minutes were thematically analysed. A pragmatic, iterative, adapted form of framework analysis was adopted [10–12].

 

Two independent researchers familiarised themselves with the qualitative data from both sources and independently defined key themes and sub-themes based on participant and SIREN research team responses. Themes were compared and agreed by consensus, and presented to other members of the team in the case of query or non-agreement. Once agreed, themes, sub-themes, and counts were summarised into a table. The research study team then collectively discussed the themes to ensure clarity of wording and agreement on the presentation of the final themes.

## Ethics statement

The SIREN study was approved by the Berkshire Research Ethics Committee (IRAS ID 284460, REC Reference 20SC0230) on 22 May 2020. The Winter Pressures sub-study was supported by two ethics amendments on 14 November 2022 and 01 December 2022. Participants were informed in advance, as the frequency and method of sampling remained the same, implied consent processes were approved by the committee. Participants returning to the study gave informed consent. Clinical trial registration number: ISRCTN11041050; registration date: 12 January 2021.

## Results

A summary of the key metrics from each pathway is available in Table 1.

## Recruitment and retention

Of the 65 sites invited to the sub-study, 39 (60.0%) opted to join, with 33 sites testing locally (Pathways A) and six sites testing at UKHSA-commissioned laboratories (Pathway B). In total, 4,289 participants were consented into the sub-study via sites (Pathways A: n = 3,713; Pathway B: n = 576).

Of the 12,549 participants invited to join the postal testing pathway (Pathway C), 3,485 (27.8%) opted to join the sub-study.

Retention across both site-based and postal pathways was high, with 4,138 (96.5%) of site-based participants, and 3,064 (87.9%) of postal participants remaining in follow-up until 31 March 2023.

Overall, 572 participants withdrew from the sub-study. The most common withdrawal reason for both site-based and postal pathways was being unable to stay in the study due to workload and/or work commitments (290/572; 50.7%) (Table 2). Compared to postal participants, a higher proportion of site-based participants reported withdrawing due to logistical issues with attending appointments (site-based: 14/152; 9.2% vs postal: 5/420; 1.2%, p < 0.001), difficulties with

**Table 1. Comparison of Pathways A, B, and C.**

|  | Pathway A | Pathway B | Pathway C |
|---|---|---|---|
| **Site recruitment** | All active sites invited, those with capacity took part | Sites interested in Pathway A, but without capacity for on-site multiplex, were offered Pathway B | No site recruitment |
| **Participant recruitment** | Participants automatically enrolled, with an option to opt-out via withdrawal | Participants automatically enrolled, with an option to opt-out via withdrawal | Participants not enrolled at a Pathway A or B site were invited to take part via an opt-in online survey |
| **PCR swabbing and testing** | PCR swabs taken on-site and tested at local hospital laboratories | PCR swabs taken on-site, but sent to UKHSA-commissioned laboratories for testing | PCR swabs taken at-home, and returned to UKHSA-commissioned laboratories for testing |
| **Multiplex testing start date** | Range: 28th November 2022–16th March 2023 | Range: 28th November 2022–7th February 2023 | Range: 22nd December 2022–24th January 2023 |
| **Median number tests per participant** | 6 (IQR: 4–7) | 4 (IQR: 3–5) | 5 (IQR: 4–6) |
| **Proportion of returned swabs that were multiplexed** | 45.6% | 100.0% | 100.0% |

**Table 2. Number of participants withdrawn by withdrawal reason and pathway.**

| Withdrawal Reason | All Participants | | Site-based participants | | Postal participants | | p-value (site-based vs postal)ᵃ |
|---|---|---|---|---|---|---|---|
| | n | % | n | % | n | % | |
| Workload and/or other work commitments | 290 | 50.7 | 66 | 43.4 | 224 | 53.3 | 0.038 |
| Moving sites/leaving the healthcare workforce | 56 | 9.8 | 24 | 15.8 | 32 | 7.6 | 0.006 |
| Frequency of testing is too high | 30 | 5.2 | 2 | 1.3 | 28 | 6.7 | 0.010 |
| Medical reasons | 25 | 4.4 | 11 | 7.2 | 14 | 3.3 | 0.061 |
| Logistical issues with attending appointments | 19 | 3.3 | 14 | 9.2 | 5 | 1.2 | <0.001 |
| Dislike the testing methods | 19 | 3.3 | 0 | 0.0 | 19 | 4.5 | 0.003 |
| Difficulties with accessing testing | 12 | 2.1 | 8 | 5.3 | 4 | 1.0 | 0.004 |
| Maternity leave/pregnancy | 4 | 0.7 | 3 | 2.0 | 1 | 0.2 | 0.059 |
| Preferred testing at site | 3 | 0.5 | 0 | 0.0 | 3 | 0.7 | 0.569 |
| Lack of results reporting | 3 | 0.5 | 0 | 0.0 | 3 | 0.7 | 0.569 |
| Impact of asymptomatic positives | 2 | 0.3 | 2 | 1.3 | 0 | 0.0 | 0.070 |
| Issues with repeat positive results | 1 | 0.2 | 0 | 0.0 | 1 | 0.2 | 1.000 |
| Other | 86 | 15.0 | 20 | 13.2 | 66 | 15.7 | 0.509 |
| Not stated | 22 | 3.8 | 2 | 1.3 | 20 | 4.8 | 0.082 |
| **Total** | **572** | **100.0** | **152** | **100.0** | **420** | **100.0** | |

*Site-based participants (Pathways A and B): Participants swabbing at hospital sites*

*Postal participants (Pathway C): Participants swabbing at home*

ᵃ*Fisher's exact test*

accessing testing (site-based: 8/152; 5.3% vs postal: 4/420; 1.0%, p = 0.004) and moving sites/leaving the healthcare workforce (site-based: 24/152; 15.8% vs postal: 32/420; 7.6%, p = −0.006). Compared to site-based participants, a higher proportion of postal participants reported withdrawing due to the frequency of testing (site-based: 2/152; 1.3% vs postal: 28/420; 6.7%, p = 0.010) and disliking the testing method (site-based: 0/152; 0.0% vs postal: 19/420; 4.5%, p = 0.003).

## Timing of introduction and coverage of multiplex testing

Between 28 November 2022 and 31 March 2023, 76.8% of participants (5,967/7,774) completed at least one SARS-CoV-2 test, and 60.9% (4,735) completed at least one multiplex test. Overall, 26,556 SARS-CoV-2 PCR tests were performed, of which 73.0% (19,396) were multiplex. Swab return rate was slightly higher for participants in Pathway C (80.5%) compared to Pathways A and B (73.7% and 73.6%, respectively, p < 0.001). The median number of tests per participant was similar across pathways (Pathway A: 6, IQR = 4–7; Pathway B: 4, IQR = 3–5; Pathway C: 5, IQR = 4–6). However, only 45.6% of PCR tests were multiplex in Pathway A, compared to 100.0% in Pathways B and C (p < 0.001) (Fig 1).

On retrospective questioning of sites in Pathway A to confirm the date of assay switchover (responses received from 13 of 33 sites), dates ranged from 28 November 2022 (which was the sub-study start date) to 16 March 2023; the median date sites switched was 15 December 2022. For Pathway B, the site start date ranged from 28 November 2022 to 7 February 2023 and all swabs sent to UKHSA-commissioned laboratories were tested on multiplex assays.

Pathway C participants were first recruited on 13 December 2022, and the first samples were taken on 22 December 2022. All samples were tested on multiplex.

## Participant survey

1,204/7,774 (15.5%) participants completed the participant research experience survey about their experience of the Winter Pressures sub-study. Of these, 556 were participants at sites (Pathways A and B have been grouped as the participant

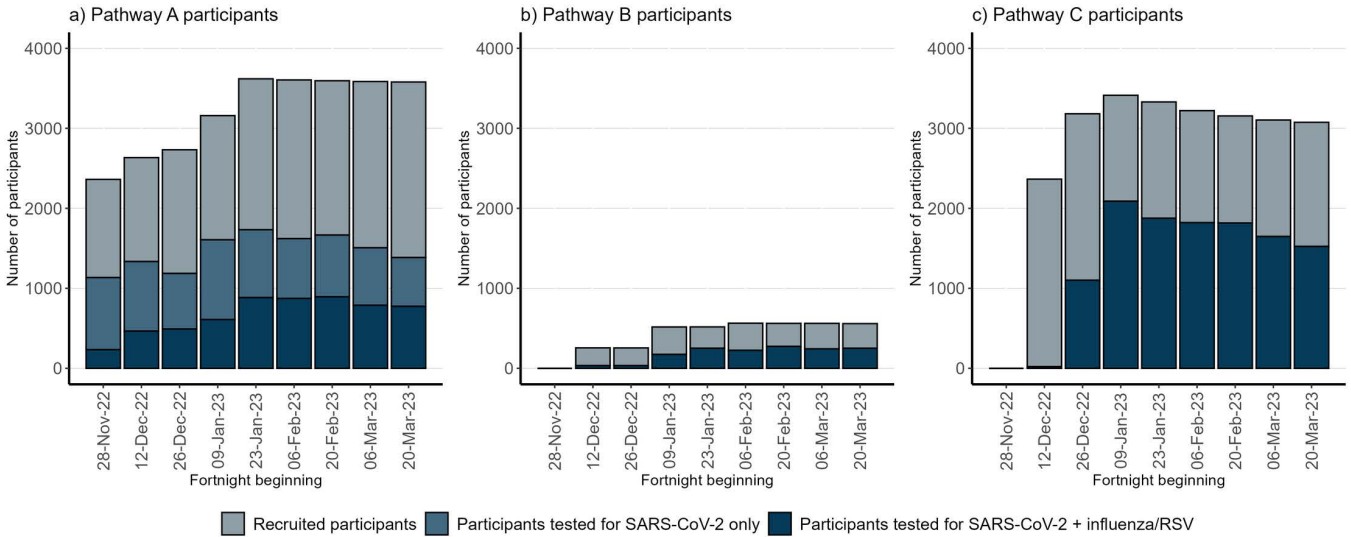

**Fig 1. Number of participants recruited and tested per fortnight, by pathway.** *"Recruited" refers to the number of participants enrolled based on site/participant start dates. Pathway A: On-site PCR swabbing, swabs analysed at local hospital laboratories Pathway B: On-site PCR swabbing, swabs analysed at UKHSA-commissioned laboratories Pathway C: At-home PCR swabbing, swabs analysed at UKHSA-commissioned laboratories.*

experience for both pathways involved site-based swabbing), 617 were postal participants (Pathway C), and 31 did not state their pathway.

A higher proportion of site-based participants agreed to feeling valued (96.0% vs 89.8%, p < 0.001), feeling respected (97.1% vs 83.9%, p < 0.001), were happy with management of their results (82.3% vs 73.0%, p < 0.001) and knowing who to contact (91.4% vs 81.9%, p < 0.001), compared to postal participants (Fig 2). Overall, 98.2% of participants agreed they would consider taking part in research again.

The three most mentioned themes in the qualitative data from the survey were 'Taking part in SIREN creates a sense of contribution', 'Logistics of the study', and 'Participant Results'. Within the 'Contribution' theme, participants felt that informing research/policy was an important factor in contributing to their participation in research. One example was:

*"If taking part helps with the good research and you gain better understanding of these viruses, I'm happy to continue".*

Within 'Participant Results', participants felt that not being informed early enough about their infection and antibody status was a disadvantage of the sub-study, compared to previous SIREN study protocols. A common suggestion was:

"*having results more frequently*".

Participants reported more logistical issues in the postal pathway regarding receiving and returning swabs compared to site-based swabbing. A challenge was familiarising themselves with new practises, i.e., posting swabs themselves. A suggested improvement was:

"*having the swabs sent out earlier and in bulk as [the] first samples were hit by a postal strike*".

Although participants did not receive incentives, this was not a deterrent to participation. For a comprehensive overview of themes and counts, see Table 3.

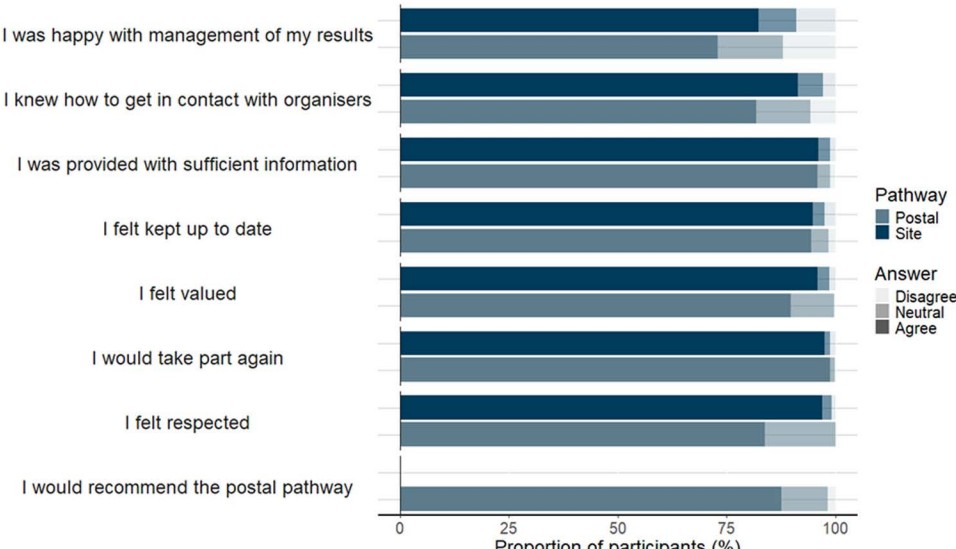

**Fig 2. Participant research experience survey feedback from participants, divided by pathway.** *Site Pathway (Pathways A and B): Participants swabbing at hospital sites. Postal Pathway (Pathway C): Participants swabbing at home.*

**Table 3. Participant research experience survey.**

| Theme | Code | Sites | Postal | Unspecified | Total |
|---|---|---|---|---|---|
| Taking part in SIREN creates a sense of contribution | Belief that taking part in SIREN contributes to society and/or research and policy | 208 | 253 | 14 | 475 |
| | Taking part in SIREN makes them feel valued | 14 | 2 | 0 | 16 |
| Participant Results | Participant knowledge of respiratory virus and antibody status | 135 | 24 | 8 | 167 |
| | Participants received no/delayed results | 24 | 86 | 1 | 111 |
| | Participants want detailed results/feedback | 26 | 4 | 0 | 30 |
| Logistics of the Study | Ease of study logistics for participants | 75 | 92 | 2 | 169 |
| | Swab/survey synchronisation | 1 | 43 | 0 | 44 |
| | Posting difficulties/convenience of sites | 0 | 29 | 0 | 29 |
| | Received unexpected number of swabs | 1 | 25 | 0 | 26 |
| | Issues travelling to site | 11 | 1 | 0 | 12 |
| | Posting more convenient than on-site testing | 0 | 10 | 1 | 11 |
| | Issues with phlebotomy | 9 | 1 | 0 | 10 |
| | Issues with swabbing | 3 | 5 | 1 | 9 |
| Communication | Friendly/efficient site teams | 75 | 6 | 3 | 84 |
| | Poor communication from SIREN | 22 | 52 | 0 | 74 |
| | Good communication from SIREN | 23 | 25 | 1 | 49 |
| | Reminders to test | 5 | 11 | 1 | 17 |
| Personal benefit | Learning about science/ results | 7 | 4 | 0 | 11 |
| | Need for incentives | 2 | 6 | 0 | 8 |
| Surveys | Issues with survey/forms | 4 | 11 | 0 | 15 |
| Wish for SIREN to continue | | 12 | 2 | 0 | 14 |

**UKHSA SIREN research team workshop**

Twenty out of twenty-one researchers attended the workshop and provided feedback on study delivery. A variety of staff groups were represented, including laboratory, data systems, clinical, operations, and leadership. Thematic analysis of the Winter Pressures sub-study workshop identified "Communications" and "Collaboration" as the most mentioned themes (Table 4). Effective internal communications were key to sub-study delivery, including using the *"monthly meeting to update the wider team"* and the importance of gaining the *"consensus of [the] whole team"*. External communications, such *as 'clear, considered participant communications'* and '*engaging the right partners at the right time'* were also important. Key challenges faced included team resourcing challenges with multiple concurrent projects leading to a '*need to prioritise'* and the need for clear set-up timelines, particularly when working with external partners, for example: *'Give sites deadlines and accept that some barriers are too difficult to overcome'.*

## Discussion

The SIREN Winter Pressures 2022−23 sub-study involved rapid adjustment of the SIREN study protocol. Modifying the multi-centre study to switch to multiplex analysis of PCR tests proved challenging to achieve at pace, so a new postal pathway was rapidly established. This evaluation provides an opportunity to reflect and compare the site-based (A and B) and postal (C) swabbing and testing pathways.

The original SIREN study, set-up during the COVID-19 pandemic, was made possible by sites joining the study and rapidly recruiting and testing participants. This evaluation shows that a postal model can have important benefits. Compared to the postal pathway, set up and delivery of site-based pathways proved time-intensive for site and central research teams and lacked standardisation, leading to more complexity, gaps in the data, and creating barriers to effective, timely data collection. The addition of the postal pathway enabled target recruitment to be achieved (7,774/7,500). However, with only 60.9% of participants testing for influenza, and the prolonged switch to multiplex resulting in testing starting after the influenza peak, the utility of data collected was affected (Table 1) [13]. This meant we were unable to conduct the planned analyses, including seasonal influenza vaccine effectiveness. The overall return rate was similar across pathways, with a slightly higher percentage of returned kits for the postal pathway compared to site-based (80.7% vs 73.7%). This is higher than a previously reported postal PCR sample return rate of 39.9% [14].

Multiplex PCR testing coverage was highly varied for sites in pathway A (testing at local laboratories), with significant differences in the date when assays were transitioned from monoplex testing (Table 1). Initial recruitment figures were based on the number of consented participants who had not withdrawn at site start date. However, actual numbers of

**Table 4. UKHSA research team workshop.**

| Themes | Sub-themes | Count |
|---|---|---|
| Communication | Communication between the UKHSA SIREN team and external stakeholders | 14 |
| | UKHSA SIREN team internal communications | 11 |
| Collaboration | Engaging the most relevant collaborators | 11 |
| | Organisational support between the SIREN team, the wider UKHSA and external partners | 9 |
| | Clear UKHSA SIREN team roles | 4 |
| Strategy and Structures | Effective planning, process design, and contingency planning | 10 |
| | Setting study aims and objectives | 6 |
| | Regular agreement and discussion within the UKHSA SIREN team | 5 |
| | Quality assurance and monitoring | 2 |
| Timescales | Includes timescales, deadlines, and parallel processes | 21 |
| Resources and Prioritisation | Includes use of wider support from UKHSA and collaborators, prioritisation, and financial processes | 15 |

tests analysed for influenza and RSV were lower than expected for participants taking part through Pathway A due to delays in local laboratories switching to multiplex testing (Table 1). In comparison, PCR tests from Pathways B and C were sent to UKHSA-commissioned laboratories for testing, where all swabs were analysed using multiplex. Variation in local laboratory multiplex testing coverage was a key limitation of data collection in the Winter Pressures sub-study, and hence to the ability to draw conclusions from the dataset. This process evaluation therefore highlights the benefits of a single, centralised laboratory for effective data collection.

This delay in sites switching to multiplex PCR testing, as in Pathway A, could be mitigated by providing longer lead-in timeframes for site study set-up and a clear deadline for switch-over. This was highlighted in the SIREN research team workshop. Feedback included the importance of engaging the right partners at the right time as crucial for set-up. However, not using sites for study delivery can reduce the time needed for study set-up, as seen in Pathway C. A shorter timeframe can reduce administrative burden, enable standardisation of testing start-date, and facilitate testing start date coinciding with the predicted peak of circulating infection levels.

A comparison of site-based (A and B) versus postal (C) pathways shows that both options were acceptable to participants. This was reflected across participant feedback, swab return rates, and retention. Participants reported positive feedback for both site-based and postal approaches. A higher proportion of participants in site-based pathways agreed to feeling valued, feeling respected, being happy with management of their results, and knowing who to contact, compared to the postal pathway. Responses to open questions suggest this may be due to participants valuing friendly and efficient site teams, highlighting the important benefits of site research teams for participant engagement. Both arms reported a sense of contribution (see Table 3). Overall, 98.2% of participants agreed they would consider taking part in research again, a higher percentage than the 92% reported by the NIHR Participants in Research Survey 2022/23 [9].

Retention was high in both site-based and postal participants, at 96.5% and 87.9% respectively, which is above the 73.9% overall mean retention reported in metanalysis of longitudinal cohort studies [15]. Participants undergoing postal testing were more likely to withdraw than those at sites, primarily due to more postal participants reporting that workload and other commitments prevented them from completing the study. Compared to participants on the postal pathway, site-based participants were more likely to withdraw due to logistical challenges. However, all participants took on average the same number of tests, suggesting this was not a major consideration for most.

Taken together, while site-based pathways showed a small benefit for participant engagement over the postal pathway, the implications of these findings are that a postal model may offer advantages to a site-based model for use in pandemic and emerging health threat scenarios. This is because the postal pathway offers greater control over the end-to-end processes and minimises the potential for delays introduced by a model that uses sites as intermediaries. For example, in the comparison between site-based and postal pathways, site-based pathways led to lower testing coverage than the postal pathway. Putting the central research team in direct contact with participants and laboratories enables rapid implementation of changes to the study, making it adaptable and, as a result, responsive to rapidly developing public health emergencies or threats.

A strength of this mixed methods evaluation of the sub-study is that it incorporates multiple sources of quantitative and qualitative data. A key limitation is that participants in the sub-study are a subset of the original SIREN cohort, and therefore may be more motivated to take part than those who withdrew or opted not to take part in the sub-study. This is particularly relevant for those on the postal pathway, who were recruited through an opt-in process. In comparison, site-based pathway participants were recruited unless they opted out. This limits generalisability to less motivated cohorts. A second limitation was that sites may have been conducting multiplex PCR testing, however the data pipeline between sites and UKHSA may have been unable to access 100% of data provided due to data linkage issues (i.e., differences in spellings of names and errors in date of birth, which were used for matching testing results to study participants).

An additional limitation is the low participant survey response rate (15.5%). Therefore, feedback may reflect the views of the most engaged participants but be unrepresentative of the overall cohort and the survey data should be interpreted

with caution. Additionally, while the SIREN research team feedback had input from all but one team member, it was collected in a team-led workshop rather than a formalised focus group moderated by independent researchers and there was no option for private feedback. However, UKHSA SIREN team debriefs have now been adapted into the team culture to support continuous improvement of study delivery.

## Conclusion

The Winter Pressures sub-study was a pragmatic adaptation of the existing SIREN protocol, which faced real-world challenges impacting data collection. While the target recruitment of participants was met, the number of returned multiplex test results did not achieve the intended large-scale seasonal coverage. Our comparison of the three pathways used, suggests that the postal pathway was the most effective format for data collection and was acceptable to participants and the research team. The postal pathway offered greater control over the end-to-end processes, providing an adaptable and responsive format. Therefore, this study suggests that a postal study model may offer advantages to a site-based model, with implications for use in pandemic preparedness and for emerging public health threats.

## Supporting information

**S1 File: Participant research experience survey questions.**
(PDF)

**S2 File: Aggregate recruitment and testing data.**
(CSV)

**S3 File: Anonymised participant research experience survey responses.**
(CSV)

## Acknowledgments

Our thanks go to the participants in the SIREN study and in particular those who took part in the Winter Pressures sub-study 2022/23, and completed the evaluation questionnaire. We would also like to thank the research team and site staff.

Lead author for the SIREN study group: Victoria Hall. Email: Victoria.Hall@ukhsa.gov.uk

SIREN study group: John Northfield (Site research team), Sean Cutler (Site research team), Anna Roynon (Site research team), Maxine Nash (Site research team), Amanda Dell (Site research team), Louise Parfitt (Site research team), Andrea Richards (Site research team), Andrea Price (Site research team), Christian Subbe (Site research team), Caroline Mulvaney Jones (Site research team), Julia Roberts (Site research team), Manny Bagary (Site research team), Nadezda Starkova (Site research team), Inderpreet Athwal (Site research team), Louise Hudson (Site research team), Ashley Jones (Site research team), Rebecca Chapman (Site research team), Lucy Booth (Site research team), Claire Williams (Site research team), Fiona Adair (Site research team), April Hawkins (Site research team), Chinari Subudhi (Site research team), Scott Latham (Site research team), Raksha Mistry (Site research team), Natalie Silva (Site research team), Abigail Severn (Site research team), Alejandro Arenas-Pinto (Site research team), Eva McAlpine (Site research team), Aran Dhillon (Site research team), Connor McAlpine (Site research team), Gosala Gopalakrishnan (Site research team), Sarah Creer (Site research team), Eve Etell Kirby (Site research team), Kim Gray (Site research team), Joanna Wright (Site research team), Joely Morgan (Site research team), Gemma Harrison (Site research team), Mark Broadhurst (Site research team), Simon Taylor (Site research team), Clare McAdam (Site research team), Natalie Crooks (Site research team), Stacey Horne (Site research team), Anna Grice (Site research team), Nicola Walker (Site research team), Luke Bedford (Site research team), Paul Ridley (Site research team), Alison O'Kelly (Site research team), Catherine Sinclair (Site research team), Val Irvine (Site research team), Elizabeth Boyd (Site research team), Claire Thomas (Site research

team), Ina Hoad (Site research team), Tryphena Konala (Site research team), Judith Radmore (Site research team), Emily Macnaughton (Site research team), Sarah Knight (Site research team), Kim Hulacka (Site research team), Robert Shorten (Site research team), Kathryn Hollinshead (Site research team), Lois Bullen (Site research team), Robert Shorten (Site research team), Claire Corless (Site research team), Sarah Mcloughlin (Site research team), Bethany Preece (Site research team), Sarah Baillon (Site research team), Samantha Hamer (Site research team), Joanne Edgar (Site research team), Kelly Moran (Site research team), Vijayendra Waykar (Site research team), Charlotte Wesson (Site research team), Rebecca Rutter (Site research team), Maureen Williams (Site research team), Bethany Jones (Site research team), Russell Coram (Site research team), Holly Slater (Site research team), Joanne Jones (Site research team), Banher Sandhu (Site research team), Elijah Matovu (Site research team), Claire Gabriel (Site research team), Katherine Pagett (Site research team), Sheron Clarke (Site research team), Sally Mavin (Site research team), Sebastien Fagegaltier (Site research team), Shannon Proctor (Site research team), Mary Summerscales (Site research team), Andrew Gibson (Site research team), Alexandra Cochrane (Site research team), Dawid Dytmer (Site research team), Lita Kovina (Site research team), Grace Davies (Site research team), Manish Patel (Site research team), Berni Welsh (Site research team), Karen Black (Site research team), Kate Templeton (Site research team), Sam Donaldson (Site research team), Andrea Clarke (Site research team), Jane Crowe (Site research team), Kadiga Campbell (Site research team), Barbara Hamilton (Site research team), Liz Sheridan (Site research team), Charlotte Barclay (Site research team), Maxine Ashton (Site research team), Alison Rodger (Site research team), Tabitha Mahungu (Site research team), Debbie Delgado (Site research team), Julia Vasant (Site research team), Deborah Howcroft (Site research team), Sarah Meisner (Site research team), Abby Rand (Site research team), Catherine Thompson (Site research team), Sophia Strong-Sheldrake (Site research team), Vicky King (Site research team), Emma Underhill (Site research team), Kate Seymour (Site research team), Holly Morgan (Site research team), Ash Turner (Site research team), Anne Hayes (Site research team), Masood Aga (Site research team), James Pethick (Site research team), Ashok Dadrah (Site research team), Thushan de Silva (Site research team), Helen Shulver (Site research team), Gareth Stephens (Site research team), Simon Tazzyman (Site research team), Mandy Carnahan (Site research team), Mandy Beekes (Site research team), Sanal Jose (Site research team), Jo stickley (Site research team), Hannah Gibson (Site research team), Yuri Protaschik (Site research team), Susan Regan (Site research team), Alison Campbell (Site research team), John Day (Site research team), Swapna Kunhunny (Site research team), Bernard Hadebe (Site research team), Paula Harman (Site research team), Sharon Tysoe (Site research team), Bridgett Masunda (Site research team), Nigara Atayeva (Site research team), Joanne Galliford (Site research team), Prisca Gondo (Site research team), Raji Orath Prabakaran (Site research team), Jane Dare (Site research team), Qi Zheng (Site research team), Danielle McCracken (Site research team), Emmanuel Defever (Site research team), Ellene Thompson (Site research team), Lynda Fothergill (Site research team), Karen Burns (Site research team), Andrew Higham (Site research team), Lisa Bishop (Site research team), Aileen Menzies (Site research team), Matt Horton (Site research team), Therese Kelly (Site research team), Cristina Dragu (Site research team), David Hilton (Site research team), Hannah Jory (Site research team), Penny Harris (Site research team), Susan Hopkins (UKHSA SIREN team), Victoria Hall (UKHSA SIREN team), Jasmin Islam (UKHSA SIREN team), Ana Atti (UKHSA SIREN team), Omoyeni Adebiyi (UKHSA SIREN team), Nick Andrews (UKHSA SIREN team), Hannah Emmett (UKHSA SIREN team), Jonathan Broad (UKHSA SIREN team), Nish Kapirial (UKHSA SIREN team), Simone Dyer (UKHSA SIREN team), Sophie Russell (UKHSA SIREN team), Colin Brown (UKHSA SIREN team), Joanna Conneely (UKHSA SIREN team), Paul Conneely (UKHSA SIREN team), Sarah Foulkes (UKHSA SIREN team), Nabila Fowles-Gutierrez (UKHSA SIREN team), Nipunadi Hettiarachchi (UKHSA SIREN team), Jameel Khawam (UKHSA SIREN team), Edward Monk (UKHSA SIREN team), Katie Munro (UKHSA SIREN team), Andrew Taylor-Kerr (UKHSA SIREN team), Jean Timeyin (UKHSA SIREN team), Edgar Wellington (UKHSA SIREN team), Angela Dunne (UKHSA SIREN team), Dominic Sparkes (UKHSA SIREN team), Naomi Platt (UKHSA SIREN team), Anna Howells (UKHSA SIREN team), Enemona Adaji (UKHSA SIREN team), Omolola Akinbami (UKHSA SIREN team), Palak Joshi (UKHSA SIREN team), Paola Barbero (UKHSA SIREN team), Meera Chand

(UKHSA SIREN team), Andre Charlett (UKHSA SIREN team), Michelle Cole (UKHSA SIREN team), Claire Neill (UKHSA SIREN team), Anne-Marie O'Connell (UKHSA SIREN team), Ferdinando Insalata (UKHSA SIREN team), Tim Brooks (UKHSA SIREN team), Maria Zambon (UKHSA SIREN team), Mary Ramsay (UKHSA SIREN team), Ayoub Saei (UKHSA SIREN team), Ezra Linley (UKHSA SIREN team), Simon Tonge (UKHSA SIREN team), Ashley Otter (UKHSA SIREN team), Silvia D'Arcangelo (UKHSA SIREN team), Cathy Rowe (UKHSA SIREN team), Amanda Semper (UKHSA SIREN team), Eileen Gallagher (UKHSA SIREN team), Robert Kyffin (UKHSA SIREN team), Kate Howell (UKHSA SIREN team), Jacqueline Hewson (UKHSA SIREN team), Iain Milligan (UKHSA SIREN team), Noshin Sajedi (UKHSA SIREN team), Davina Calbraith (UKHSA SIREN team), Caio Tranquillini (UKHSA SIREN team), Jerry Ye Aung Kyaw (UKHSA SIREN team), Lisa Cromey (Public Health Agency Northern Ireland), Dianne Corrigan (Public Health Agency Northern Ireland), Desmond Areghan (Glasgow Caledonian University), Jennifer Bishop (Public Health Scotland), Melanie Dembinsky (Glasgow Caledonian University), Laura Dobbie (Public Health Scotland), Josie Evans (Public Health Scotland), David Goldberg (Public Health Scotland), Lynne Haahr (Glasgow Caledonian University & Public Health Scotland), Annelysse Jorgenson (Glasgow Caledonian University), Ayodeji Matuluko (Glasgow Caledonian University), Laura Naismith (Public Health Scotland), Desy Nuryunarsih (Glasgow Caledonian University & Public Health Scotland), Alexander Olaoye (Glasgow Caledonian University), Caitlin Plank (Public Health Scotland), Lesley Price (Glasgow Caledonian University & Public Health Scotland), Nicole Sergenson (Glasgow Caledonian University & Public Health Scotland), Sally Stewart (Glasgow Caledonian University & Public Health Scotland), Andrew Telfer (Public Health Scotland), Jennifer Weir (Public Health Scotland), Ellen De Lacy (Public Health Wales), Yvette Ellis (Health and Care Research Wales), Susannah Froude (Public Health Wales), Chris Norman (Health and Care Research Wales), Guy Stevens (Public Health Wales), Linda Tyson (Public Health Wales).

## Author contributions

**Conceptualization:** Sophie Russell, Victoria Hall.

**Data curation:** Katie Munro.

**Formal analysis:** Sophie Russell, Katie Munro, Sarah Foulkes, Jonathan Broad.

**Funding acquisition:** Victoria Hall.

**Investigation:** Katie Munro.

**Methodology:** Sophie Russell, Katie Munro, Sarah Foulkes, Jonathan Broad, Dominic Sparkes, Ana Atti, Jasmin Islam, Victoria Hall.

**Project administration:** Ana Atti, Jasmin Islam, Victoria Hall.

**Supervision:** Sarah Foulkes, Susan Hopkins, Victoria Hall.

**Validation:** Katie Munro, Sarah Foulkes.

**Visualization:** Sophie Russell, Katie Munro, Sarah Foulkes.

**Writing – original draft:** Sophie Russell, Katie Munro, Sarah Foulkes, Jonathan Broad, Dominic Sparkes, Victoria Hall.

**Writing – review & editing:** Sophie Russell, Katie Munro, Sarah Foulkes, Jonathan Broad, Dominic Sparkes, Ana Atti, Jasmin Islam, Susan Hopkins, Victoria Hall.

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
