## [Decision Letter · Decision Letter 0]

23 Jan 2025

Dear Dr. Munro,

Thank you for submitting your manuscript to PLOS ONE. After careful consideration, we feel that it has merit but does not fully meet PLOS ONE’s publication criteria as it currently stands. Therefore, we invite you to submit a revised version of the manuscript that addresses the points raised during the review process.

We look forward to receiving your revised manuscript.

Kind regards,

Farhana Haque, MBBS MPH MSc PhD

Academic Editor

PLOS ONE

Journal Requirements:

[The SIREN study was funded by the UK Health Security Agency; the UK Department of Health and Social Care with contributions from the governments in Northern Ireland, Wales, and Scotland; the National Institute for Health Research; Health Data Research UK (NIHR200927; HDRUK2022.0322)].

3. In the online submission form, you indicated that Anonymised data will be made available for secondary analysis to trusted researchers upon reasonable request.].

4. One of the noted authors is a group or consortium [SIREN study group]. In addition to naming the author group, please list the individual authors and affiliations within this group in the acknowledgments section of your manuscript. Please also indicate clearly a lead author for this group along with a contact email address.

6. We note that there is identifying data in the Supporting Information file <S2_File.pdf>. Due to the inclusion of these potentially identifying data, we have removed this file from your file inventory. Prior to sharing human research participant data, authors should consult with an ethics committee to ensure data are shared in accordance with participant consent and all applicable local laws.

-Location data

Additional guidance on preparing raw data for publication can be found in our Data Policy (https://journals.plos.org/plosone/s/data-availability#loc-human-research-participant-data-and-other-sensitive-data ) and in the following article: http://www.bmj.com/content/340/bmj.c181.long .

Please remove or anonymize all personal information (Names), ensure that the data shared are in accordance with participant consent, and re-upload a fully anonymized data set. Please note that spreadsheet columns with personal information must be removed and not hidden as all hidden columns will appear in the published file.

Reviewers' comments:

Reviewer's Responses to Questions

**Comments to the Author**

1. Is the manuscript technically sound, and do the data support the conclusions?

Reviewer #1: Yes

Reviewer #2: Yes

2. Has the statistical analysis been performed appropriately and rigorously?

Reviewer #1: Yes

Reviewer #2: Yes

3. Have the authors made all data underlying the findings in their manuscript fully available?

Reviewer #1: No

Reviewer #2: Yes

4. Is the manuscript presented in an intelligible fashion and written in standard English?

Reviewer #1: Yes

Reviewer #2: Yes

Reviewer #1: Thank you for allowing me to review "Adapting COVID-19 research infrastructure to capture influenza and RSV alongside SARS-CoV-2 in UK healthcare workers winter 2022/23: Evaluation of the SIREN Winter Pressures pilot study".

The paper was a well designed and written evaluation of the Winter Pressures study and with minor changes outlined below should be published.

Line 35 – I’d like to know the breakdown of the participants for pathway A vs B – it is good to know that 33 sites used pathway A (for example) but how many people was that? Especially since for pathway C you have to report participant number (as there are no “sites”)

Line 38 – Instead of 6; 4; 5 in the brackets for median number of test (which is I assume is what that means), you could put median 5, range 4-6, unless it is important that pathway A had 6, etc.

Line 163 – it might be my misunderstanding but does the “without limit” refer to the content of the free text response (how it could be interpreted as written now) or the length (character limit) of the response?

Line 186 – qualitative is spelled incorrectly.

Line 190 – “The research study team then discussed collectively to finalise themes”

Line 217 – I find it a bit confusing that you’ve switched the order of the comparison for the brackets in this section. I think keep site first and postal second in all comparison brackets. Or some other way (than looking at the denominator) to make it clear which numbers belong to which category.

Table 2 – it is interesting that the site based participants reported medical reasons more than postal participants too – think about adding that to the wording.

Table 2 – can you do any statistical comparison to see if the differences between sites are significant?

Line 228 – it would also be good to see a statistical comparison for the swab return rate.

Line 240 – There were 33 sites in pathway A. You say that they switched to multiplex at various dates between 28 November to 16 March. Given this is a large proportion of the participants and it impacts on the ability to identify flu and RSV, it would be useful to know how many each week – given there were still a lot of people tested only for SARS-CoV-2 in the last time point in the figure, it seems that quite a lot of the 33 sites didn’t switch until very late. For all we can tell from the text at the moment, 1 might have switched on 28 November and the other 32 on 16 March! (exaggeration as I see median is in December) I’m not sure if this would be best in a table or figure, and if it should be in the body of the paper or supplemental but I know I would like to know!

Line 365 – did you provide workshop participants with an option to provide feedback privately? Not all people like to speak opinions in front of a group.

Reviewer #2: This manuscript presents a relevant study on adapting research infrastructure during the winter of 2022/23 to capture multiple respiratory viruses in UK healthcare workers. However, there are several areas that need improvement before it can be considered for publication.

1. Lack of clarity and structure: The introduction could do a better job of clearly stating the overall project, research design, and aims of the sub-study. It seems a bit muddled, please make it clear from the start what the key research question and why use this process evaluation. Second, some of the paragraphs are a bit short in length, it would benefit from clearer delineation between sections such as Background, Methods, Results, and Conclusion. This could enhance readability and allow us for easier navigation.

2. Sample representativeness: There's not enough information about whether the recruited participants and sites are representative of the broader population of UK healthcare workers. How were the sites selected? Were there any biases in the recruitment process? here’s less information.

3. Data collection issues: methods for data collection, particularly for participant feedback, should be elaborated upon. What specific questions were asked in the feedback survey? How was thematic analysis conducted?

4. Data analysis: The results section presents numerical data but lacks sufficient statistical analysis. For example, it mentions that the median number of tests per participant was similar across pathways but does not provide statistical tests (e.g., ANOVA or Kruskal-Wallis test) to support this claim. Also Including confidence intervals for key metrics would strengthen the findings by providing a measure of precision.

5. Informed Consent: More detail on how informed consent was obtained from participants would be beneficial, especially regarding those re-recruited into the sub-study. And the ethical review process and any relevant ethical considerations specific to this study seems missing.

6. Low survey response rate: With only 15.5% of participants completing the feedback survey, the results might be highly skewed and not reflective of the whole cohort. Please try to either find a way to increase the response rate in future research or be more cautious when interpreting the survey data.

7. Discussions: the limitation could consider, for instance, how might biases in participant selection or feedback affect the results? Additionally, how does the variation in local laboratory testing impact overall study conclusions?

8. Implications of Findings: The authors should elaborate on what the results mean for future public health strategies regarding respiratory virus testing, particularly in light of potential future pandemics.

**Do you want your identity to be public for this peer review?** For information about this choice, including consent withdrawal, please see our Privacy Policy

Reviewer #1: No

Reviewer #2: No

---

## [Author Response · Author response to Decision Letter 1]

10 Mar 2025

Response to reviewers has been included in the documents ("Response to Reviewers")

---

## [Decision Letter · Decision Letter 1]

9 May 2025

Dear Dr. Munro,

Thank you for submitting your manuscript to PLOS ONE. After careful consideration, we feel that it has merit but does not fully meet PLOS ONE’s publication criteria as it currently stands. Therefore, we invite you to submit a revised version of the manuscript that addresses the points raised during the review process.

**Please kindly address the few minor comments from Reviewer 1 and resubmit the manuscript. **

We look forward to receiving your revised manuscript.

Kind regards,

Farhana Haque, MBBS MPH MSc PhD

Academic Editor

PLOS ONE

**Journal Requirements:**

Reviewers' comments:

Reviewer's Responses to Questions

**Comments to the Author**

Reviewer #1: All comments have been addressed

Reviewer #2: All comments have been addressed

2. Is the manuscript technically sound, and do the data support the conclusions?

Reviewer #1: Yes

Reviewer #2: Yes

3. Has the statistical analysis been performed appropriately and rigorously?

Reviewer #1: I Don't Know

Reviewer #2: Yes

4. Have the authors made all data underlying the findings in their manuscript fully available?

Reviewer #1: Yes

Reviewer #2: Yes

5. Is the manuscript presented in an intelligible fashion and written in standard English?

Reviewer #1: Yes

Reviewer #2: Yes

**Reviewer #1: ** Thank you for addressing the comments. The paper looks great. I particularly like Table 1 - this makes it really clear, and the addition of stats elsewhere in the paper really support the story.

Could the p values be added to Table 2? This would also help with interpretation. At the very least, it would be good to be able to identify the statistically significant comparisons via an asterisk.

The formatting of some of the cross references has caused some of the Table references to drop to the next line. This is usually because an "enter" has got caught up in the table legend. This may be cleaned up in desktopping though.

**Reviewer #2:**  I appreciate the authors' revisions to the manuscript and their detailed response to prior reviewer comments. The revised version of the paper titled "Adapting COVID-19 research infrastructure to capture influenza and RSV alongside SARS-CoV-2 in UK healthcare workers winter 2022/23: Evaluation of the SIREN Winter Pressures pilot study" has improved considerably in clarity and structure. The integration of mixed methods and the focus on implementation pathways in real-world settings provide insights for future public health surveillance studies.

**Do you want your identity to be public for this peer review?** For information about this choice, including consent withdrawal, please see our Privacy Policy

Reviewer #1: No

Reviewer #2: No

---

## [Author Response · Author response to Decision Letter 2]

14 May 2025

Response to reviewers can be found in the uploaded "Response to reviewers" document.

---

## [Editor Report · Decision Letter 2]

28 May 2025

Adapting COVID-19 research infrastructure to capture influenza and RSV alongside SARS-CoV-2 in UK healthcare workers winter 2022/23: Evaluation of the SIREN Winter Pressures pilot study

PONE-D-24-39117R2

Dear Dr. Munro,

We’re pleased to inform you that your manuscript has been judged scientifically suitable for publication and will be formally accepted for publication once it meets all outstanding technical requirements.

Kind regards,

Farhana Haque, MBBS MPH MSc PhD

Academic Editor

PLOS ONE
---

## [Editor Report · Acceptance letter]

PONE-D-24-39117R2

PLOS ONE

Dear Dr. Munro,

I'm pleased to inform you that your manuscript has been deemed suitable for publication in PLOS ONE. Congratulations! Your manuscript is now being handed over to our production team.

Kind regards,

on behalf of

Dr Farhana Haque

Academic Editor

PLOS ONE